# Structural basis of HIV-1 Vif-mediated E3 ligase targeting of host APOBEC3H

Fumiaki Ito [1,2,3], Ana L. Alvarez-Cabrera[2,3], Kyumin Kim [1], Z. Hong Zhou[2,3] & Xiaojiang S. Chen [1,4,5,6] ✉

Human APOBEC3 (A3) cytidine deaminases are antiviral factors that are particularly potent against retroviruses. As a countermeasure, HIV-1 uses a viral infectivity factor (Vif) to target specific human A3s for proteasomal degradation. Vif recruits cellular transcription cofactor CBF-β and Cullin-5 (CUL5) RING E3 ubiquitin ligase to bind different A3s distinctively, but how this is accomplished remains unclear in the absence of the atomic structure of the complex. Here, we present the cryo-EM structures of HIV-1 Vif in complex with human A3H, CBF-β and components of CUL5 ubiquitin ligase (CUL5, ELOB, and ELOC). Vif nucleates the entire complex by directly binding four human proteins, A3H, CBF-β, CUL5, and ELOC. The structures reveal a large interface area between A3H and Vif, primarily mediated by an α-helical side of A3H and a five-stranded β-sheet of Vif. This A3H-Vif interface unveils the basis for sensitivity-modulating polymorphism of both proteins, including a previously reported gain-of-function mutation in Vif isolated from HIV/AIDS patients. Our structural and functional results provide insights into the remarkable interplay between HIV and humans and would inform development efforts for anti-HIV therapeutics.

The human innate immune system possesses a diversified repertoire of APOBEC3 (A3) restriction factors against viruses[1,2]. The interplay between viral infectivity factor (Vif) and host A3 cytidine deaminases represents a hallmark of the front-line molecular battle between HIV and host cell immunity. Among seven known members of the A3 subfamily, A3F, A3G, and A3H possess potent restriction activity against lentiviruses, including Vif-deficient HIV-1. These A3 members are stealthily packaged into budding HIV virions and trigger C-to-U mutation on the cDNA reverse-transcript, leading to G-to-A lethal hypermutation on the positive strand proviral genome, therefore effectively limiting viral proliferation[3–7]. As a remarkable tit-for-tat example between viruses and hosts, HIV deploys Vif, a protein encoded by all lentiviruses known to date, to counteract A3-mediated virus restriction[8–11]. In this process, Vif adopts a "double-hijack" mechanism

by recruiting cellular T-cell transcription cofactor CBF-β and Cullin-5 (CUL5) E3 ubiquitin ligase complex comprising CUL5, Elogin B (ELOB) and C (ELOC), and RING-box protein 2 (RBX2) to induce polyubiquitin chain formation on A3 proteins for subsequent proteasomal degradation[12–14].

Human A3H is the sole member of the single-domain A3 subfamily that shows potent anti-HIV activity[7,15]. A3H is packaged into HIV virions when stably expressed in T-cells and deaminates cytosines on the negative strand of the cDNA HIV genome with a preference for a 5′-TC sequence context, resulting in GA-to-AA mutations in the integrated proviral genomes[7,16,17]. A3H associates with cellular and viral RNA with high affinity, which is implicated in virion packaging, catalytic activity regulation, higher-order oligomerization, and sub-cellular localization[18,19]. Recent crystal structures of A3H from humans,

[1]Molecular and Computational Biology, Department of Biological Sciences, University of Southern California, Los Angeles, CA 90089, USA. [2]Department of Microbiology, Immunology and Molecular Genetics, University of California, Los Angeles CA90095, USA. [3]California NanoSystems Institute, University of California, Los Angeles CA90095, USA. [4]Genetic, Molecular and Cellular Biology Program, Keck School of Medicine, University of Southern California, Los Angeles CA90089, USA. [5]Norris Comprehensive Cancer Center, University of Southern California, Los Angeles CA90089, USA. [6]Center of Excellence in NanoBiophysics, University of Southern California, Los Angeles CA90089, USA. ✉e-mail: xiaojiac@usc.edu

chimpanzee, and pig-tailed macaque showed unique double-stranded RNA (dsRNA) binding, which leads to an unusual RNA-mediated dimer[20–22]. RNA binding can be disrupted by introducing mutations in the RNA-binding loop, a phenomenon that was harnessed to determine the structure of an RNA-free form of human A3H[19].

The remarkable characteristic of Vif is its versatility in antagonizing multiple A3 members. The fact that Vif uses at least three distinct surface areas to bind different A3s poses a challenge to unveiling the mechanistic basis of their molecular interactions[1,2]. HIV-1 Vif has strain-specific polymorphisms at amino acid residue position 48, an observation that can be correlated with its ability to antagonize A3H and the infectivity of HIV[11,23]. Furthermore, Vif polymorphism at position 63 observed in HIV clones isolated from either HIV patients or lab-adaptation studies has been linked to their ability to antagonize a specific A3H haplotype and reflects an endless molecular arms race between HIV and humans[24,25]. However, how these multiple interacting interfaces contribute to the remarkable interplay between HIV and host remains incomplete due to the lack of high-resolution structures of Vif-host factor complexes.

We report here the cryo-EM structures of human A3H bound to HIV-1 Vif in complex with CBF-β and multiple components of CUL5 E3 ligase. The structures reveal the architecture of the multicomponent HIV-human protein complex and illuminate the molecular details of the crucial interface between A3H and Vif that enables stable polyubiquitin chain formation and degradation of A3H. Our study further provides insights into the A3H ubiquitination sites targeted by the Vif-CUL5 E3 ligase complex.

## Results

### Reconstitution of a functional A3H-Vif complex in vitro

To reconstitute the A3H-Vif complex, we explored constructs of A3H and Vif that can stably form a protein complex in vitro. Vif proteins from different HIV-1 strains display varying levels of antagonism against human A3H. For example, Vif from a reference strain LAI antagonizes A3H more effectively than that from another reference strain NL4-3, and thus A3H is more restrictive against NL4-3 than against LAI. This difference in antagonistic activity of Vif has been attributed to a single Vif amino acid polymorphism at position 48, histidine and asparagine in LAI and NL4-3, respectively (Fig. 1a)[11,23]. On the other hand, studies that compared cross-species specificity of A3H have identified that A3H amino acid at position 97 modulates sensitivity to Vif of both HIV-1 and simian immunodeficiency virus from chimpanzee (SIVcpz). Human A3H possesses lysine at this position (K97) and is largely resistant to Vif from a subset of reference HIV-1 strains, including NL4-3, and fully resistant to Vif from SIVcpz; by contrast, A3H from chimpanzee (cpzA3H) possesses glutamine at the same position (Q97) and is sensitive to Vif from a wide range of HIV-1 and SIVcpz strains (Fig. 1b)[26,27]. Vif-mediated degradation assay showed that the addition of N48H point-mutation in NL4-3 Vif substantially lowered the steady-state level of A3H when Vif and A3H were co-transfected in HEK293T cells (Fig. 1c). Moreover, K97Q point-mutation in human A3H resulted in a reduced level of A3H when co-transfected with WT NL4-3 Vif, and complete depletion of A3H when co-transfected with N48H NL4-3 Vif (Fig. 1c).

We next tested in vitro complex formation using purified human A3H and HIV-1 Vif complexed with human CBF-β/ELOB/ELOC/CUL5/RBX2 (referred to as "VCBCCR" complex hereafter) (Fig. 1d). Of note, the majority of A3H was purified in RNA-bound form, which is consistent with previous crystal structures of A3Hs from human, chimpanzee, and pig-tailed macaque[20–22]. In agreement with the Vif-mediated degradation assay, K97Q point-mutation in A3H caused a predominant peak shift in the elution profiles of Superdex 200 size-exclusion chromatography (SEC) when mixed with VCBCCR containing either WT or N48H NL4-3 Vif. The combination of K97Q A3H and N48H Vif exhibited the most significant peak shift with at least two

obvious complex species, suggesting that the A3H·VCBCC complex exists in multiple stoichiometry or oligomeric states (Fig. 1d). The complex species with the largest peak shift from the mixture of K97Q A3H and N48H Vif (equivalent to peak 1 in the bottom left SEC profile in Fig. 1d) were isolated and used for the cryo-EM study.

In vitro ubiquitination assay showed essentially the same trend, as the addition of K97Q mutation enhanced the rate of mono- and polyubiquitination of A3H, and the most efficient ubiquitination pattern was observed when K97Q A3H was combined with N48H Vif (Fig. 1e). Altogether, these results highlight the determinants of molecular interaction between A3H and Vif and demonstrate that the resulting protein complex is functional in vitro.

### Architecture of A3H·VCBCC complex

Next, we analyzed the in vitro assembled A3H·Vif complex by single-particle cryo-EM. To minimize domain motion of the elongated CUL5, only the N-terminal domain was included in the CUL5 construct (CUL5-NTD) (Fig. 2a). The assembled A3H/Vif/CBF-β/ELOB/ELOC/CUL5 (A3H-VCBCC) complex was resolved to a resolution of 3.2 Å (Fig. 2b, Supplementary Figs. 1–3, Supplementary Table 1, and Supplementary Movie 1). The overall complex has a butterfly-like shape with four lobes (Fig. 2b, Supplementary Fig. 1), formed respectively by A3H/RNA, Vif/CBF-β, ELOB/ELOC, and CUL5-NTD. A3H is located near the α/β-domain of Vif and features a typical cytidine deaminase fold with a five-stranded β-sheet core surrounded by six α-helices (Fig. 2b). A3H contacts Vif through its α2-α3-α4 three-helix bundle, which contains the zinc-coordinating cytidine deaminase active site. On the other hand, Vif contacts A3H with its concave side formed by the distorted five-stranded antiparallel β-sheet (β2-β6). No major contacts are observed between A3H and the rest of the VCBCC complex, except for a charge-charge interaction between A3H and CBF-β (discussed later). A clear feature of an A-form dsRNA is observed above the odd-numbered loops of A3H (i.e., loops 1, 3, 5, and 7) and extends away from the core of the complex. Different from previous crystal structures[20–22], the second A3H molecule across the dsRNA was only partially present as it is only visible at a low-density threshold level in the cryo-EM map, possibly due to occasional dissociation or relative flexibility of the second A3H subunit.

The architecture of the VCBCC complex core resembles that of the previously reported crystal structure[28]. Vif α/β-domain forms a globular core with CBF-β by forming an antiparallel β-sheet between Vif β1 and CBF-β β2-3. A small Vif α-domain with a zinc-coordinating site mediates the recruitment of ELOB/C and CUL5 through a three-way interaction between Vif-ELOC-CUL5, which is typical for CUL5 recruiting SOCS-box containing substrate receptors[29,30]. For CUL5, only the first Cullin-repeat (CR1) was visible among the three repeats that consist of the N-terminal arm domain. The same cryo-EM data contained a class of A3H-unbound dimeric form of VCBCC complex, which was reconstructed to 3.5 Å resolution (Supplementary Fig. 1). The VCBCC complex dimerization is mainly mediated by CBF-β, which closely resembles one of the units of the crystal packing of the VCBCC complex within its asymmetric unit (PDB ID: 4N9F)[28]. All three Cullin-repeats (CR1-3) are visible in this form, likely because the Cullin-arm is stabilized by the additional contact with Vif from the other molecule within the dimer (Supplementary Fig. 4a, b). The CR1 of CUL5 exhibits about 10 degrees of rotation compared to the crystal structure when Vif/CBF-β core is superimposed, indicative of a certain extent of conformational flexibility of the Cullin-arm (Supplementary Fig. 4c).

### A3H-Vif primary interface

A3H and Vif form an extensive interface with a total buried surface of 643 Å². The protein-protein interaction is mediated by a combination of hydrophilic and hydrophobic networks primarily between an area around α3 and α4 of A3H and the five-stranded β-sheet (β2-6) of the α/β domain of Vif (Fig. 3a and Supplementary Movie 2).

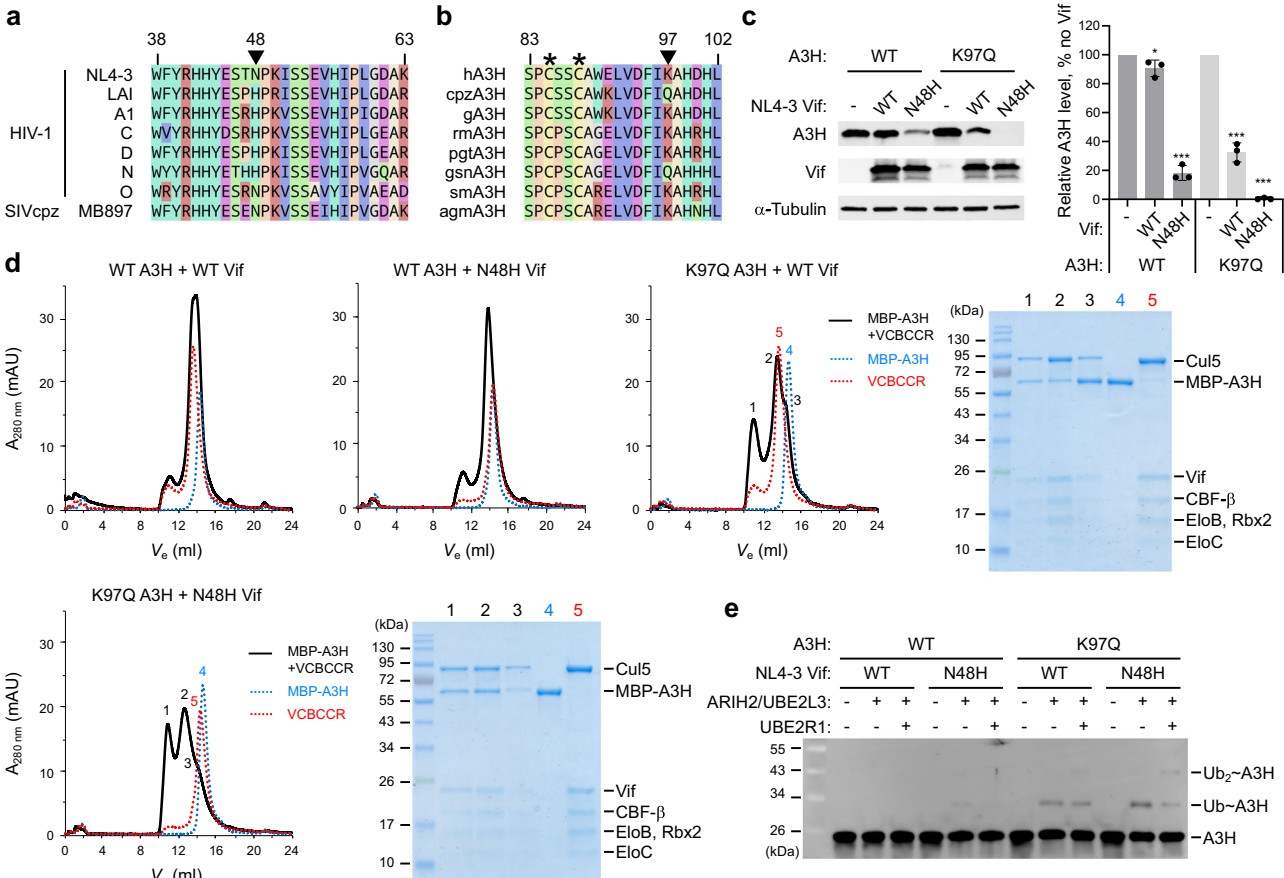

**Fig. 1 | Reconstitution of functional A3H-Vif complex. a** Alignment of Vif sequences from various HIV-1/SIV strains. At amino acid position 48 (marked by a triangle), NL4-3 Vif has asparagine (N48), whereas LAI Vif has histidine (H48). **b** Alignment of primate A3H sequences. At amino acid position 97 (marked by a triangle), human A3H has lysine (K97), whereas chimpanzee A3H has glutamine (Q97). Conserved zinc-coordinating cysteines are marked by asterisks. hA3H, human; cpzA3H, chimpanzee; gA3H, gorilla; rmA3H, rhesus macaque; pgtA3H, pig-tailed macaque; gsnA3H, golden snub-nosed monkey; smA3H, sooty mangabey; agmA3G, African green monkey. **c** Vif-mediated degradation assay of A3H in HEK293T cells. A3H and Vif plasmids were co-transfected, and steady-state levels of A3H at post-48 h transfection were analyzed by Western blots. Quantified A3H levels are shown in the right-side chart. Assays were performed independently in triplicate (mean ± s.d.; $n = 3$). The statistical significance of A3H degradation was assessed by two-tailed t-test assuming equal variance; *$P < 0.05$; **$P < 0.005$;

***$P < 0.0005$; NS, not significant. *P*-values are 0.0477, <0.0001, <0.0001, and <0.0001 for WT A3H + WT Vif, WT A3H + N48H Vif, K97Q A3H + WT Vif, and K97Q A3H + N48H Vif, respectively. The addition of N48H mutation in Vif lowered the levels of both WT and K97Q A3H. K97Q mutation in A3H lowered the A3H levels when transfected with either WT or N48H Vif. An anti-α-tubulin antibody was used as a loading control. **d** Binding analysis of A3H and VCBCCR complex by SEC. The biggest peak shift was observed when K97Q A3H was combined with N48H NL4-3 Vif. The fractions corresponding to peak 1 were used for the cryo-EM study. MBP-fused A3H was used in this assay. SDS-PAGE gels show the protein components in the indicated SEC fractions. **e** In vitro ubiquitination assays. Vif-mediated ubiquitin chain initiation and elongation on the purified A3H were tested with or without ubiquitin-conjugating enzymes ARIH2/UBE2L3 and UBE2R1, respectively. Mono-ubiquitination and poly-ubiquitination were the most efficient when the K97Q A3H was combined with N48H NL4-3 Vif. Source data are provided as a Source Data file.

Among the structural elements involved in the interface, the A3H four-turn helix α3 is central to the interaction as it is entirely buried within the interface with Vif. A3H S86 between the two zinc-coordinating cysteines (C85 and C88) forms a hydrogen bond with E45 of Vif loop 3. W90 and D94 interact with amphipathic R41 of Vif β2; an indole ring of W90 forms a hydrophobic interaction with the aliphatic chain of R41, while the side-chain carboxyl of D94 forms an electrostatic interaction with guanidine moiety of R41. Interestingly, the Vif-sensitivity determining residue Q97 of A3H is indeed located in the A3H-Vif interface. Q97 forms an intermolecular hydrogen bond with K63 of Vif β3 while being stabilized by an intramolecular hydrogen bond with the S129 from the neighboring α4 of A3H (Fig. 3b). A3H α4 expands the interface with Vif. A3H haplotype-specific residue D121[31,32] forms a hydrogen bond with the hydroxyl group of Y30 at the end of Vif α1. L125 forms a hydrophobic interaction with F39 of Vif β2 (Fig. 3c). Outside the A3H α3 and α4, two additional electrostatic interactions are observed between A3H D100 and Vif K92, and A3H E70 and Vif R93 (Fig. 3d).

In total, nine residues of A3H and eight residues of Vif are involved in the A3H-Vif interaction (Fig. 3e); no obvious interaction was observed between either of the RNA strands and Vif. Vif-mediated degradation assay using A3H mutants on the residues involved in the Vif-binding showed that the tested mutations rendered A3H more resistant to Vif with varying degrees. Among the α3 and α4 mutants tested, W90A rendered the A3H fully resistant to Vif. D94A, L125R, and S129R mutations also rendered the A3H highly resistant, increasing the A3H levels by nearly 50% in the presence of Vif (Fig. 3f). We then tested the roles of the two electrostatic bonds outside the α3-α4 patch of A3H. A charge-reversing mutation E70R on loop 4 rendered the A3H partially resistant (~40%). When combined with another charge-reversing mutation in the neighboring residue D100 on loop 6, the double-mutant E70R/D100R became fully resistant to Vif (Fig. 3g). We further validated the roles of these electrostatic interactions by mutating Vif. Like the results of the A3H mutants, R93E Vif showed reduced capacity to degrade A3H, increasing the A3H level by 70%. The double mutant K92E/R93E Vif showed essentially no degradation of

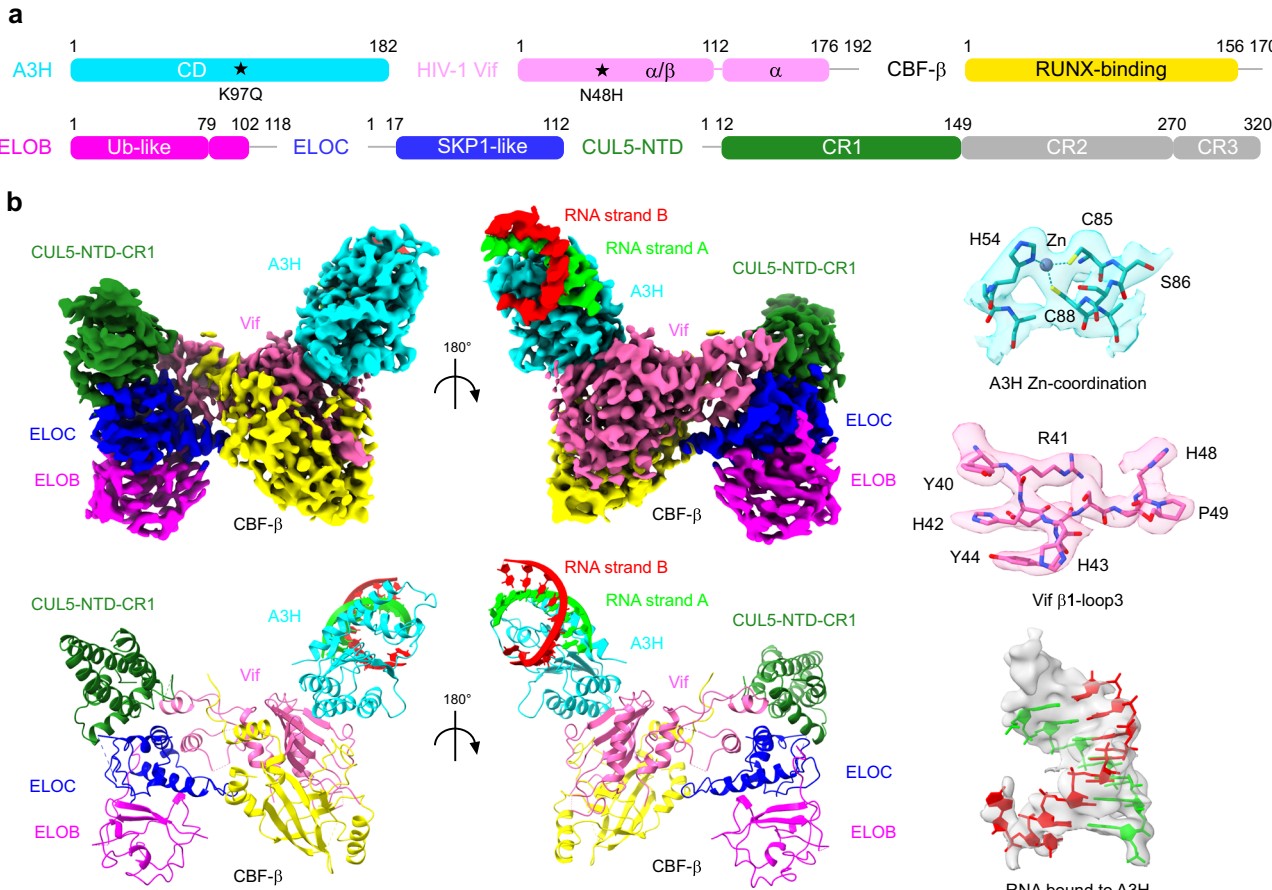

**Fig. 2 | Cryo-EM structure of the A3H-VCBCC complex. a** Domain organization and construct design of A3H and VCBCC complex. CD cytidine deaminase, CR Cullin repeat. Gray lines are not included in the construct. In CUL5-NTD, the invisible regions in the cryo-EM map are colored in gray. The key point-mutations for the stable complex assembly are marked with stars. **b** 3.2 Å cryo-EM reconstruction (top) and resultant atomic model (bottom) of the A3H-VCBCC complex. Representative cryo-EM densities of A3H, Vif, and RNA are shown in semi-transparent density superposed with corresponding atomic models (sticks).

A3H (Fig. 3h). Of note, swapping the charges between A3H and Vif by using A3H E70R−Vif R93E pair and A3H E70R/D100R−Vif K92E/R93E pair did not restore the degradation of A3H, indicating that other interactions or minor differences in the orientation of the amino acid side chains may prevent the restoration of the interaction between A3H and Vif (Supplementary Fig. 5). These results indicate that while the α3 and α4 of A3H are major structural components of Vif-interface, the two electrostatic interactions outside the α3-α4 patch are also essential to the A3H-Vif interaction. The results also indicate that the A3H E70−Vif R93 pair plays a more dominant role than A3H D100−Vif K92 pair.

**A3H lysine residues for Vif-mediated ubiquitination**
When substrate proteins for proteasomal degradation are ubiquitinated, they typically accept ubiquitin molecules through isopeptide bonds between the ε-amino group of one or multiple lysine residues on the surface of the substrate proteins and C-terminal end carboxyl group of ubiquitin. Our in vitro ubiquitination assay suggested that A3H is ubiquitinated in a Vif-dependent manner (Fig. 1e). A3H undergoes ubiquitination in vivo as well when Vif is present in cells[33]. To gain an insight into the mechanism of Vif-mediated ubiquitination of A3H, we obtained a 5.1 Å cryo-EM reconstruction of the A3H-VCBCCR complex, which contains the C-terminal domain of CUL5 (CUL5-CTD) and RBX2 that together mediate the recruitment of the ubiquitin-loaded E2 ubiquitin-conjugating enzymes (Supplementary Fig. 6). The atomic model fitting into the low-resolution density map revealed that a module comprising CUL5-CTD and RBX2 (C/R module) points toward

A3H bound to Vif. The spatial gap between the C/R module and A3H is ~40 Å, which is expected to be occupied with E2 ubiquitin-conjugating enzymes such as ARIH2/UBE2L3 pair and UBE2R1 for ubiquitin chain initiation and elongation, respectively (Supplementary Fig. 7)[34–36].

In search for the surface residues on A3H that accept ubiquitin molecules, we found that a surface area around loops 1 and 3 containing clustered lysine residues apparently faces the direction of the C/R module where ubiquitin-conjugating enzymes would bind (Fig. 4a). Loop 3 possesses a lysine-rich patch including three consecutive lysines -⁵⁰KKK⁵²-, and this lysine-rich sequence is widely conserved across primate A3H (Fig. 4b, c). There is an additional conserved surface-exposed lysine in the neighboring loop 1 (K27) (Fig. 4b, c). Vif-mediated degradation assay showed that while either the mutations on the three consecutive lysines on loop 3 (K50R/K51R/K52R) or one on loop 1 (K27R) showed only a marginal increase in the resistance to Vif, the combination of these mutations rendered A3H largely resistant to Vif (Fig. 4d). These results indicate that Vif-CUL5 ligase complex ubiquitinates a range of reachable lysine residues around loops 1 and 3 of A3H, rather than targeting a specific residue for ubiquitin chain formation.

**Comparison with A3G-Vif complex**
The structure of the A3H-Vif complex exhibits marked differences from the recently reported structures of the A3G-Vif complex[37–39]. Comparison of Vif-binding surfaces of A3G and A3H in these complexes shows the involvement of distinct structural elements within the conserved cytidine deaminase fold (Fig. 5a). In addition, a key

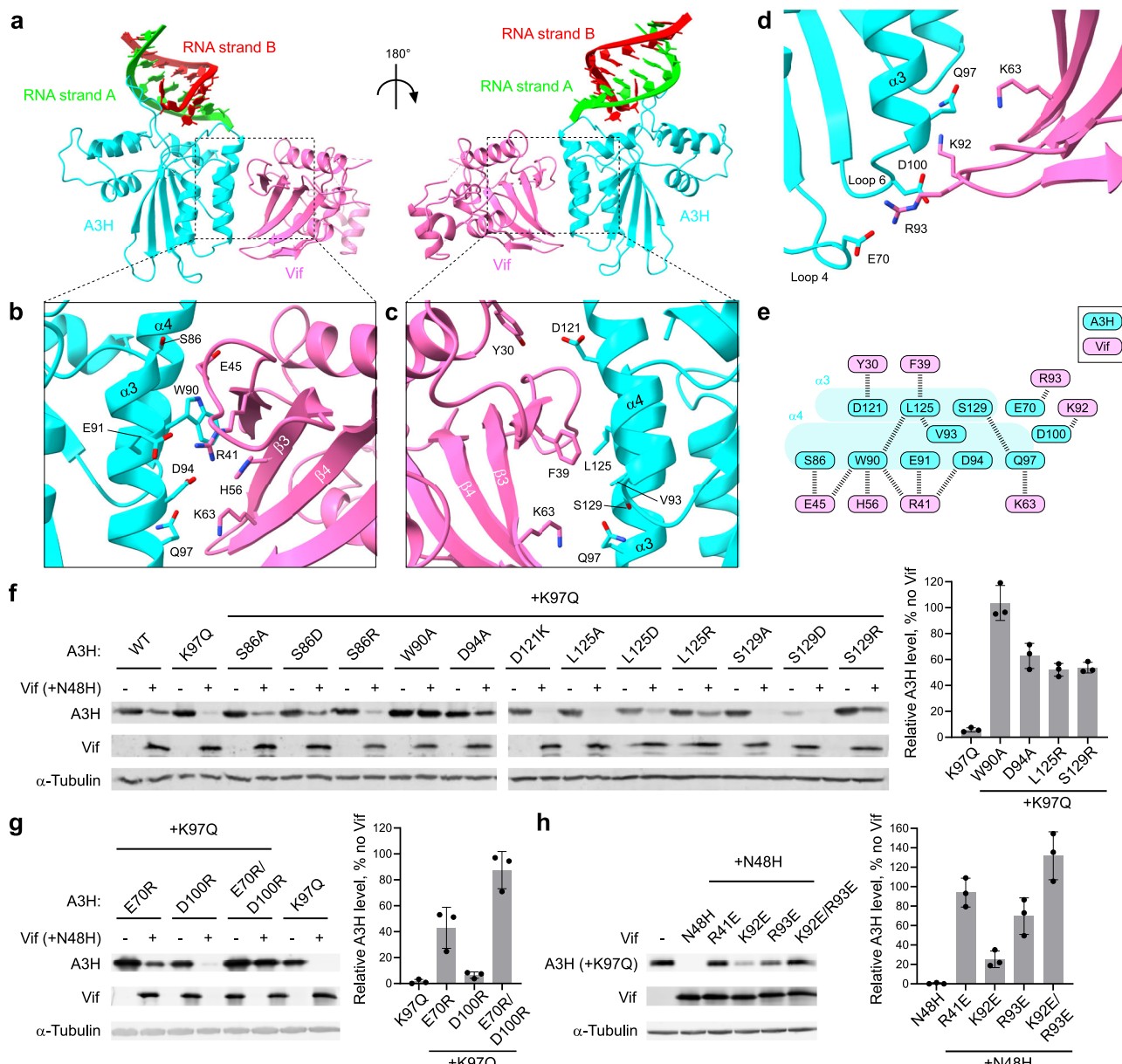

**Fig. 3 | A3H-Vif interface. a** Structure of the A3H-Vif subregion in the A3H-VCBCC complex. **b** Close-up view of the interface between A3H α3 and Vif. **c** Close-up view of the interface between A3H α4 and Vif. **d** Interface between A3H loops 4 and 6 and Vif. **e** Schematic of the amino acid residues involved in the A3H-Vif interactions. **f, g** Vif-mediated degradation assay of A3H mutants. Mutations were introduced in the Vif-interface residues around α3 and α4 (**f**) and around loops 4 and 6 (**g**) on top of K97Q Vif-sensitivity enhancing mutation. The A3H levels in the presence and absence of Vif (+N48H) were probed by Western blots. **h** Vif-mediated degradation assay using Vif mutants. Mutations were introduced in the A3H-interface residues on top of N48H. A3H levels in the presence of Vif mutants were probed by Western blots. Quantified A3H levels are shown in the right chart. Assays were performed independently in triplicate (mean ± s.d.; *n* = 3). Source data are provided as a Source Data file.

difference in the Vif-binding mode lies in the participation of bound RNA strands. In the A3G-Vif complex, the ssRNA overhang strand in the bound RNA partly mediates the A3G-Vif interaction resulting in the unique three-way interaction between A3G, Vif, and RNA. By contrast, no RNA strands are directly involved in the A3H-Vif interface. The direct protein-protein contact in the A3G-Vif complex resides in the small patch of loop7-α4 of CD1, including -128DPD130- Vif-binding motif and a single residue from CD2 (K270), while a more extensive area around the α3 and α4 consists of the Vif-binding surface in A3H. In both A3G-Vif and A3H-Vif complexes, the nature of the interactions is the combination of hydrophilic and hydrophobic networks. The different binding modes to Vif lead to the distinct surface areas for Vif-E3 ligase-mediated poly-ubiquitination: in A3G, surface lysine residues in

α3 and β5 of CD2 accept ubiquitin molecules while; but in A3H, clustered lysine residues around loops 1 and 3 accept ubiquitin molecules.

A3G and A3H share similar RNA binding regions (Fig. 5a). For example, the key structural components for RNA binding reside in loops 1, 5, 7, and α6 for both A3G and A3H. However, the structure of bound RNA and the resulting RNA recognition mechanisms differ dramatically. A3G recruits branched dsRNA by extensively interacting with a four-nucleotide-long 3'-overhang strand. On the other hand, the structure of the A3H-RNA subregion within the A3H-VCBCC complex showed that a short dsRNA (~8 bp) is branched above loop 7, leaving a two-nucleotide-long 5'-overhang RNA strand toward loop 1 and no observed 3'-overhang strand. Interestingly, the conserved aromatic patch in loop 7, i.e., -124YYFW127- in A3G and -112YYHW115- in A3H, engages

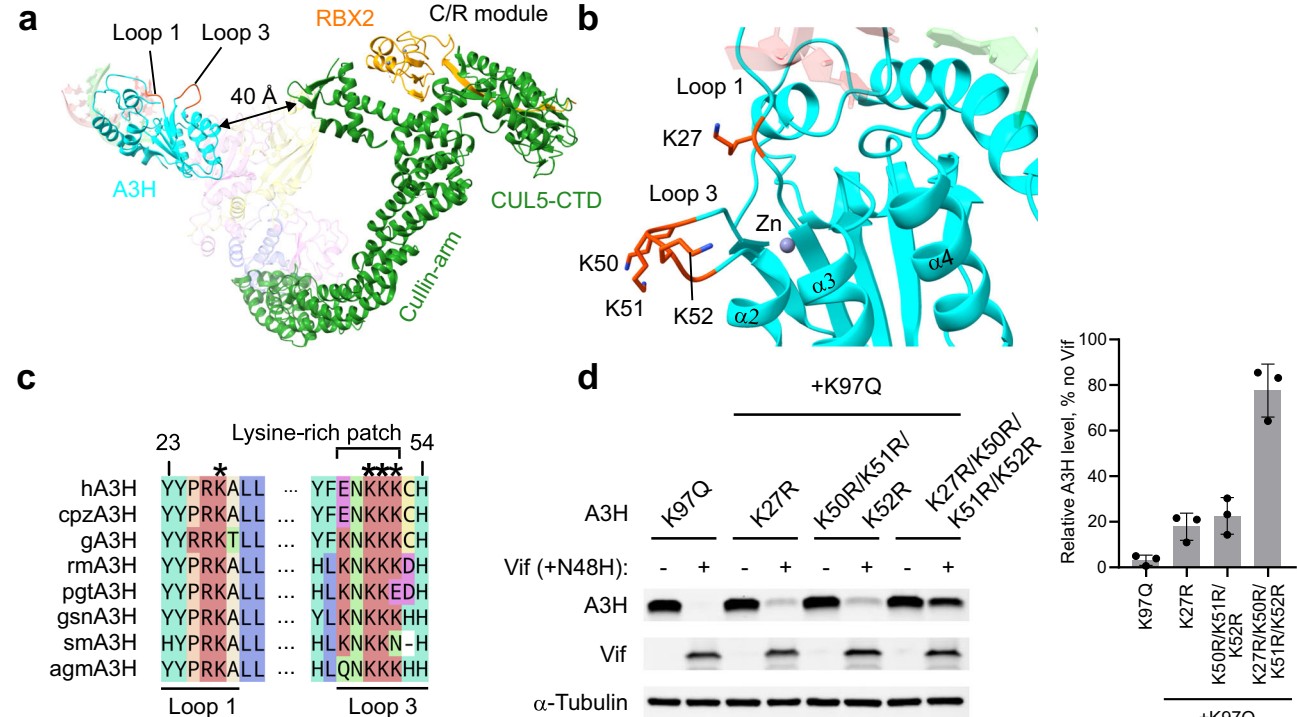

**Fig. 4 | Ubiquitination sites in A3H. a** Spatial relationship of A3H and C/R module for ubiquitin transfer based on the fitted atomic models of A3H-VCBCCR complex. **b** Clustered surface-exposed lysine residues around loops 1 and 3 of A3H. **c** Sequence alignment of primate A3H highlighting the conserved lysine residues in loops 1 and 3 that contain a lysine-rich patch. Numbers above the sequence correspond to residue numbering in hA3H. **d** Vif-mediated degradation assay of A3H lysine mutants. Lysines in loops 1 and 3 were replaced with arginines, which would maintain the positively charged side chains but block ubiquitination. The combination of four lysine-to-arginine mutations rendered A3H largely resistant to Vif-mediated degradation. The mutations were introduced on top of the K97Q Vif-sensitivity enhancing mutation. The A3H levels in the presence and absence of Vif (+N48H) were probed by Western blots. Quantified A3H levels are shown next to the gel image. Assays were performed independently in triplicate (mean ± s.d.; $n = 3$). Source data are provided as a Source Data file.

in RNA in a completely different fashion. The aromatic patch of A3G exclusively interacts with the first two nucleotides of the 3′-overhang strand. The aromatic patch of A3H interacts with the base pair immediately before the branch point as well as the first base of the 5′-overhang strand. The RNA binding of A3H in the A3H-VCBCC complex closely resembles the previously reported A3H-RNA crystal structures[20–22], indicating that no major conformational change in RNA is induced upon the binding to Vif.

Two complex structures enabled us to map Vif amino acid residues for both A3G-RNA and A3H binding (Fig. 5b). A3G-RNA-binding residues largely reside around the α1 and loop 3, while A3H-binding residues reside around the 5-stranded β-sheets and loop 3. Y30 at the end of α1 is the only residue directly involved in both complex formations. Vif loop 3 is also central to both A3G-RNA and A3H binding, while no Vif loop 3 residues are directly involved in both A3G and A3H binding.

## Discussion

Evolutionary pressure has shaped a multifaceted Vif to antagonize multiple host A3s to evade their virus restriction activity. The primate A3 has diversified to combat invading lentiviruses and other viral pathogens. In humans, the diversification resulted in seven members of the A3 proteins, each with one or two cytidine deaminase domain(s), unique RNA binding properties, and distinct target sequence preferences for cytidine deamination. Human A3H is a potent HIV restriction factor with a single cytidine deaminase domain that is classified as the unique Z3-subtype[40,41]. A3H is under active evolution, and there are at least 12 haplotypes in the human population[42]. Different haplotypes display varying degrees of protein stability, deaminase activity, antiviral activity, and sensitivity to Vif-mediated

degradation. For example, haplotype II (used in this study) is stable in cells, highly active as a cytidine deaminase, restrictive against HIV-1, and partly sensitive to Vif-mediated degradation. This is in stark contrast to haplotype I, which is unstable in cells, less active as cytidine deaminase, less restrictive against HIV-1, and fully resistant to Vif[24,31,43,44].

Our cryo-EM structure of the A3H-VCBCC complex revealed an interface that is primarily mediated by an area around α3-α4 helices of A3H with the five-stranded β-sheet of Vif, which is distinct from the binding surfaces for A3G or A3F[37–39,45]. The observed interface areas are in good agreement with the previously predicted interface[26,46]. For example, Ooms et al.[46]. and Nakashima et al.[26]. both demonstrated that A3H residues on α3 and α4 contribute to the sensitivity to Vif in cell-based degradation assays. Ooms et al. further generated a structural model by using anchor points revealed by the Vif adaptation experiments. For example, they demonstrated that Vif mutation E45Q can restore the degradation of Vif-resistant A3H mutant S86E. The cryo-EM structure we obtained here largely agrees with the predicted model. Notably, we did observe a polar interaction between A3H S86 and Vif E45 in the structure.

Vif-mediated degradation assay using A3H mutants revealed the differential contribution of the residues on α3 and α4 to the Vif-binding, with W90, D94, L125, and S129 being central to the interaction. Two electrostatic interactions outside the α3-α4 patch also critically contribute to the Vif-A3H interaction. To assemble the protein complex in vitro, we leveraged the effect of Vif-sensitivity modulating residue at 97, where human A3H naturally has a lysine residue. Consistent with its reported role[26,27], the addition of K97Q mutation in A3H facilitated the formation of the functional A3H-Vif-CUL5 E3 ligase complex. The mutated residue is, in fact, observed in the Vif-interface

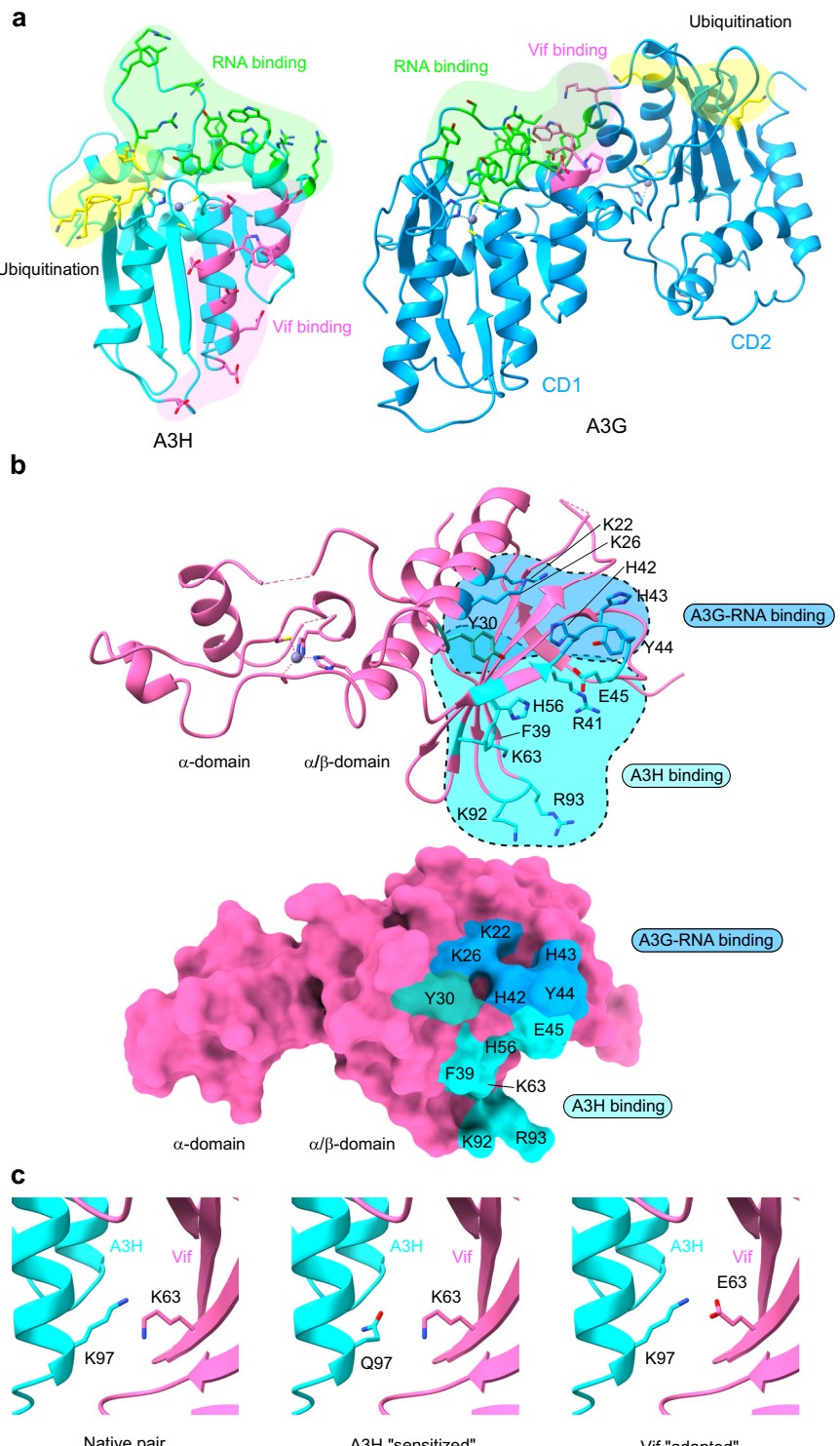

**Fig. 5 | Comparison of A3H·Vif and A3G·Vif complex and insights into determinants that modulate A3H·Vif interaction. a** Comparison of surface areas responsible for Vif binding, RNA binding, and ubiquitination in A3H (left) and A3G (right). **b** Vif surface residues responsible for A3H binding (sticks in cyan) and A3G-RNA binding (sticks in blue). Y30 is the only residue involved in both complexes (stick in pink). **c** Atomic model showing key amino-acid contacts modulating the A3H·Vif interaction. In native human A3H and HIV-1 Vif pair, A3H K97 is in close proximity to Vif K63, resulting in unfavorable contact (left). In sensitized A3H with K97Q mutation, the glutamine forms a hydrogen bond with Vif K63, thus stabilizing the interaction (middle). Alternatively, in a gain-of-function HIV strain that has adapted to A3H-mediated HIV restriction, Vif E63 can form an electrostatic interaction with K97, thus stabilizing the interaction (right).

as the glutamine forms a hydrogen bond with Vif K63 (Figs. 3b and 5c). Additional Vif residues around this location, such as R90, K92, and R93, create a positively charged surface which may disfavor K97 of human A3H (Supplementary Fig. 8 and Supplementary Movie 3). This may explain why none of the human A3H haplotypes are fully susceptible to Vif, in contrast to the highly sensitive human A3G[11,31,32]. It has been reported that Vif variants isolated from either HIV patients with homozygous haplotype II A3H or lab-adaptation study of HIV in supT11 cells expressing A3H gained a K63E charge-reversing mutation on Vif[24,25]. Modeling the K63E mutation in our A3H·Vif complex

structure indicates a favorable electrostatic interaction with K97 of human A3H (Fig. 5c). Indeed, these studies showed that HIV clones with K63E in Vif resulted in dramatically elevated infectivity and enhanced replication kinetics in the presence of A3H expression, suggesting more effective inactivation of A3H antiviral activity. Therefore, this gain-of-function mutation may represent a possible evolutionary route for the current HIV strains.

Another Vif polymorphic residue at position 48, which also modulates the antagonism of A3H[11,23], is located at the turn of loop 3. Intriguingly, the H48 is not in direct contact with A3H in the A3H·Vif complex structure, indicating that the effect of this residue in A3H-binding is rather indirect (Supplementary Fig. 9a). Moreover, Vif loop 3 superimposes well with that in the crystal structure of VCBCC complex possessing N48 in Vif, suggesting that N48H change does not induce conformational rearrangement of loop 3 that possesses A3H binding residues (Supplementary Fig. 9b). The bulkier H48 (vs. N48) is tightly packed with the CBF-β F69 and W73 as well as the neighboring Vif residue P49 (Supplementary Fig. 9c). CBF-β F69 and W73 are near and within a loop that has a long disordered sequence (amino acids 74-82) in the VCBCC complex (Supplementary Fig. 9d). H48 would partly stabilize this CBF-β loop to allow a charge-charge interaction between CBF-β E76/Q77 and A3H K64, thus facilitating the formation of the A3H·VCBCC complex (Supplementary Fig. 9e).

Vif-mediated antagonism of A3H is conserved in a wide range of SIV strains from Old world monkeys[7,40,47]. Among the Vif-binding residues identified in this study, S86, W90, and D100 in human A3H are variable in Old world monkeys (Supplementary Fig. 10). Notably, the critical α4 residue W90 is either glycine or arginine in Old world monkey A3Hs. Notwithstanding the molecular details uncovered here for the interface between human A3H and HIV Vif, further work is desirable to unveil how Vif-mediated antagonism of A3H is accomplished in other primates and their cognate SIV strains.

In summary, our structural and biochemical data provide molecular details into the complex assembly of Vif-E3 ligase and A3H, a process leading to the ubiquitination of A3H and its subsequent degradation. Interestingly, the structure captures a complex of Vif-E3 ligase binding to a form of A3H with RNA that could be packaged into virions to exert its antiviral activity. This complex structure likely represents one of the intermediate steps where HIV-1 Vif antagonizes the host immunity by specifically interfering with the A3H form that is ready for HIV virion packaging (Supplementary Fig. 11). The atomic and chemical details uncovered in this study would be helpful for future HIV therapeutic efforts targeting this fundamental virus-host interface.

## Methods
### Plasmids
Vif from HIV-1 pNL4-3 (GenBank: AF324493.2, residues 1–176) with His$_6$-tag at N-terminus, human ELOB (NCBI Reference Sequence: NP_009039.1, residues 1–102), human ELOC (NCBI Reference Sequence: NP_001191790.1, residues 17–112), and human CBF-β (NCBI Reference Sequence: NP_001746.1, residues 1–156) were cloned into each of the four polycistronic cassettes in pST39 co-expression vector[48]. The N-terminal domain of human CUL5 (NCBI Reference Sequence: NP_003469.2, residues 12–320) was cloned into pGEX-6P-1 to express a fusion protein with GST at N-terminus. Human A3H haplotype II (GenBank: ACK77776.1) was cloned into pMAL-c5X to express a fusion protein with MBP at N-terminus. Human RBX2 (NCBI Reference Sequence: NP_055060.1, residues 1–113) with His$_6$-tag at N-terminus and near-full-length CUL5 (residues 12–780) fused with GB1 at N-terminus were cloned into pETDuet-1 co-expression vector. For in vitro ubiquitination assay, HA-tag was inserted at the C-terminal end of MBP-A3H. For mammalian cell expression, FLAG-A3H, HA-A3H, and HIV-1 pNL4-3 Vif were cloned into the pcDNA3.1(+) mammalian expression vector. Cloning and mutagenesis were performed with In-Fusion cloning and PrimeSTAR mutagenesis (Clontech) by following the manufacturer's instructions. The sequences of all the constructs were verified by Sanger DNA sequencing (Azenta Life Sciences). The multiple sequence alignments were generated with Linnaeo (https://github.com/beowulfey/linnaeo).

### Vif-mediated degradation assay
FLAG-A3H-pcDNA or HA-A3H-pcDNA variants were co-transfected with either Vif-pcDNA or pcDNA3.1(+) empty vector into HEK293T cells (ATCC) by using X-tremeGENE 9 DNA Transfection Reagent (Roche). At post-48 h transfection, the cells were washed once with PBS and lysed in RIPA buffer with 1× complete protease inhibitors (Roche). The lysates were then subjected to Western blot with anti-FLAG M2 mAb from mouse (Sigma-Aldrich, 1:3000 dilution), anti-HA mAb from mouse (Sigma-Aldrich, 1:3000 dilution), anti-α-tubulin mAb from mouse (GeneTex, 1:5000) and anti-Vif mAb from mouse (NIH AIDS Reagent Program #319, 1:2000 dilution) as primary antibodies. Cy3-labeled goat-anti-mouse mAb (GE Healthcare, 1:3000 dilution) was used as a secondary antibody to detect the signal. The fluorescent signal was detected and visualized with Typhoon RGB Biomolecular Imager (GE Healthcare).

### Protein expression and purification
His$_6$-Vif/ELOB/ELOC/CBF-β-pST-39, CUL5-NTD-pGEX-6P-1, and His$_6$-RBX2/GB1-CUL5-pETDuet-1 expression vectors were transformed into the E. coli strains BL21(DE3), and A3H-pMAL-c5X expression vector was transformed into C43 (DE3) pLysS. The E. coli cells harboring the expression vectors were grown in an LB medium at 37 °C until the OD$_{600}$ reached 0.6. The recombinant proteins were induced by 0.3 mM isopropyl β-D-1-thiogalactopyranoside (IPTG) at 16 °C for 18 h.

For Vif/CBF-β/ELOB/ELOC (VCBC) complex, the cell pellets were resuspended with buffer A (20 mM Tris-HCl (pH 8.0), 500 mM NaCl, and 0.5 mM TCEP) containing RNase A (0.1 mg/ml, Qiagen), lysed by sonication, and cellular debris was removed by centrifugation. The supernatant containing the His$_6$-VCBC complex was loaded onto the Ni-NTA agarose column (Qiagen). The nickel column was extensively washed with wash buffer (20 mM Tris-HCl (pH 8.0), 500 mM NaCl, 50 mM imidazole, and 0.5 mM TCEP), and the protein was eluted with elution buffer (20 mM Tris-HCl (pH 8.0), 500 mM NaCl, 500 mM imidazole, and 0.5 mM TCEP). The His$_6$-tag was cleaved by incubating with PreScission protease overnight. VCBC complex was subjected to HiLoad 16/600 Superdex 200 pg column (Cytiva) equilibrated with buffer A. The peak fractions were collected and concentrated for mixing with either CUL5-NTD or CUL5/RBX2 complex.

For CUL5-NTD, the cell pellets were resuspended with buffer A, lysed by sonication, and cellular debris was removed by centrifugation. The supernatant containing GST-CUL5-NTD was loaded onto glutathione sepharose column (GE Healthcare). The glutathione column was extensively washed with buffer A and the GST tag was cleaved with PreScission protease on the column overnight. CUL5-NTD was eluted from the glutathione column and subjected to HiLoad 16/600 Superdex 75 pg column (GE Healthcare) equilibrated with buffer A. The peak fractions were collected and concentrated for mixing with the VCBC complex.

For CUL5/RBX2 complex, the cell pellets were resuspended with buffer A, lysed by sonication, and cellular debris was removed by centrifugation. The supernatant containing His$_6$-RBX2/GB1-CUL5 was loaded onto the Ni-NTA agarose column. The nickel column was extensively washed with wash buffer (20 mM Tris-HCl (pH 8.0), 500 mM NaCl, 50 mM imidazole, and 0.5 mM TCEP), and the protein was eluted with elution buffer (20 mM Tris-HCl (pH 8.0), 500 mM NaCl, 500 mM imidazole, and 0.5 mM TCEP). The His$_6$-tag and GB1-tag were cleaved by incubating with PreScission protease

overnight. CUL5/RBX2 complex was subjected to Superdex 200 Increase 10/300 GL column (Cytiva) equilibrated with buffer A. The peak fractions were collected and concentrated for mixing with the VCBC complex.

For MBP-A3H, the cell pellets were resuspended with the buffer B (25 mM HEPES-NaOH (pH 7.5), 500 mM NaCl, and 0.5 mM TCEP) containing RNase A (0.1 mg/ml), lysed by sonication, and cellular debris was removed by centrifugation. The supernatant containing MBP-A3H was loaded onto amylose column (New England Biolabs). The amylose column was extensively washed with wash buffer (25 mM HEPES-NaOH (pH 7.5), 1 M NaCl, and 0.5 mM TCEP), and the protein was eluted with the elution buffer (25 mM HEPES-NaOH (pH 7.5), 500 mM NaCl, 20 mM D-maltose, and 0.5 mM TCEP). Eluted fractions were concentrated and subjected to HiLoad 16/600 Superdex 200 pg column (Cytiva) equilibrated with buffer B. Peak fractions of dimeric MBP-A3H were separated and concentrated.

VCBCC complex and VCBCCR complexes were formed by mixing VCBC complex and CUL5-NTD or CUL5/RBX2 by 1:1 molar ratio and incubating on ice for 30 min. The protein complex was purified by Superdex 200 Increase 10/300 GL column. Prior to forming the complex, MBP-tag was cleaved from A3H with PreScission protease. A3H-VCBCC and A3H-VCBCCR complexes were formed by mixing them at a 1:1 molar ratio and incubating on ice for 30 min. The protein mix was further purified by Superdex 200 Increase 10/300 GL column. The peak fraction was isolated and concentrated for cryo-EM work.

ARIH2, UBE2L3 and UBE2R1 proteins for in vitro ubiquitination assay were purified as previously described[37]. Protein purity and stoichiometry of the protein complexes were assessed by SDS-PAGE at each purification step.

### SEC binding analysis
Purified MBP-A3H and VCBCCR complex were mixed by 1:1 molar ratio in binding buffer (25 mM HEPES-NaOH (pH 7.5), 300 mM NaCl, and 0.5 mM TCEP) and incubated at 4 °C for 30 min. The mixture was then subjected to Superdex 200 Increase 10/300 GL column equilibrated with the binding buffer. Fractions were concentrated and visualized by SDS-PAGE.

### In vitro ubiquitination assay
Prior to the ubiquitination assay, NEDD8 was conjugated to CUL5 in the VCBCCR complex with either WT or N48H Vif to activate the E3 ubiquitin ligase for recruiting E2 ubiquitin-conjugating enzymes. The NEDD8 conjugation reaction was performed by mixing 10 μM VCBCCR complex, 0.8 μM E1 NEDD8 activating enzyme (Enzo Life Sciences), 1 μM UBE2F (Enzo Life Sciences), 40 μM NEDD8 (Enzo Life Sciences), 15 U/ml inorganic pyrophosphatase from baker's yeast (Sigma-Aldrich), 50 mM Tris (pH 7.5), 250 mM NaCl, 5 mM MgCl$_2$, 5 mM ATP, 1 mM DTT and incubating for 2.5 h at room temperature. NEDD8-conjugated VCBCCR (N8-VCBCCR) complex was purified by Superdex 200 Increase 10/300 GL column to remove excess amount of NEDD8 and other enzymes in the reaction.

MBP-tag was cleaved from A3H with PreScission protease before the in vitro ubiquitination assay. The mono- and poly-ubiquitination of A3H were performed by mixing 1 μM HA-A3H (and its variants), 0.4 μM E1 ubiquitin-activating enzyme (Enzo Life Sciences), 0.3 μM ARIH2, 1.8 μM UBE2L3, 3.6 μM UBE2R1 (for poly-ubiquitination only), 1.5 μM N8-VCBCCR complex, 12 μM ubiquitin (Sigma-Aldrich), 15 U/ml inorganic pyrophosphatase from baker's yeast (Sigma-Aldrich), 50 mM Tris (pH 7.5), 250 mM NaCl, 5 mM MgCl$_2$, 5 mM ATP, 1 mM DTT in 20 μl reaction volume and incubating for 2 h at room temperature. The reaction was stopped by adding 2x sample buffer. The ubiquitinated products were then subjected to western blot with anti-HA mAb from mouse (Sigma-Aldrich, 1:3000 dilution) as a primary antibody. Cy3-labeled goat-anti-mouse mAb (GE Healthcare, 1:3000 dilution) was used as a secondary antibody to detect the signal. The fluorescent

signal was detected and visualized with Typhoon RGB Biomolecular Imager (GE Healthcare).

### Negative-stain EM
For this, 5 μl of 0.02 mg/ml A3H-VCBCC complex sample was applied onto glow-discharged ultrathin formvar/carbon supported copper 400-mesh grids (EMS), blotted and stained with 2.0% uranyl acetate. Negative-stained grids were imaged on a Tecnai F20 transmission electron microscope (FEI) operated at 200 kV.

### Cryo-EM data acquisition
The freshly reconstituted protein complex was lightly crosslinked with 1 mM bis-sulfosuccinimidyl suberate (BS3, Thermo Fisher Scientific) on ice for 30 min. The reaction was quenched by 50 mM Tris (pH 8.0) for an additional 10 min. Then, 4 μl aliquots of 0.15 mg/ml purified A3H-VCBCC complex or A3H-VCBCCR complex were applied to graphene oxide-coated Quantifoil R1.2/1.3 gold 400-mesh grids (Electron Microscopy Sciences). Grids were then blotted and vitrified in liquid ethane using Vitrobot Mark IV (Thermo Fisher Scientific). Vitrified EM grids were screened in Talos F200C (Thermo Fisher Scientific) or Tecnai F20 (FEI) transmission electron microscopes to optimize the freezing conditions.

Cryo-EM data of the A3H-VCBCC complex were collected in Titan Krios (Thermo Fisher Scientific) equipped with K3 direct electron detector and post-BioQuantum GIF energy filter (Gatan) operated at 300 kV in electron counting mode. Movies were collected at a nominal magnification of 165,000× and a pixel size of 0.51 Å. A total dose of 50 e$^-$/Å$^2$ per movie was used with a total exposure time of approximately 3.5 s. A total of 14,725 movies were recorded by automated data acquisition with EPU (Thermo Fisher Scientific). Cryo-EM data of the A3H-VCBCCR complex were collected in Glacios (Thermo Fisher Scientific) equipped with Falcon-4 direct electron detector operated at 200 kV in electron counting mode. Movies were collected at a nominal magnification of 150,000× and a pixel size of 0.92 Å in EER format. A total dose of 40 e$^-$/Å$^2$ per movie was used with a dose rate of 5-6 e$^-$/Å$^2$/s. A total of 12,546 movies were recorded by automated data acquisition with EPU.

### Cryo-EM data processing
The movies were imported into cryoSPARC software package[49] and subjected to patch motion correction and CTF estimation in cryoSPARC. Reference-free manual particle picking in a small subset of data was performed to generate 2D templates for auto-picking. For the A3H-VCBCC complex, a total of 2,515,035 particles were picked initially, extracted, and down-sampled by a factor of 4, on which rounds of 2D classification were performed. A total of 163,697 particles from 2D classes with clear feature were selected. To retrieve overlooked particles from the micrographs, Topaz, a convolutional neural network-based particle-picking program[50], was trained with the clean particles from the 2D classification. A total of 3,637,921 particles were extracted by Topaz and used for the second round of 2D classification. A total of 384,384 good particles were obtained, re-extracted with a down-sampling factor of 2. 3D ab initio reconstruction was then performed to generate three classes. Classes containing a dimer form of VCBCC complex and an A3H-VCBCC complex were identified. For VCBCC complex dimer, the particles were re-extracted with full resolution and non-uniform refinement[51] was performed with C2 symmetry imposed, which yielded a 3.5 Å resolution map. For the A3H-VCBCC complex, an extra round of ab initio reconstruction and heterogenous refinement was performed to yield three classes. Two good classes from the heterogenous refinement were selected, and the particles were re-extracted with full resolution. Non-uniform refinement was then performed to yield the final 3.2 Å resolution map.

For the A3H-VCBCCR complex, a total of 5,032,127 particles were picked initially, extracted, and down-sampled by a factor of 4, on which

rounds of 2D classification were performed. A total of 572,753 particles from 2D classes with clear feature were obtained and re-extracted with full resolution. 3D ab initio reconstruction was then performed to generate four classes. Two classes with apparent feature of Cullin-arm were selected. Another round of ab initio reconstruction and heterogenous refinement was performed to yield four classes. A single dominant class was selected, and non-uniform refinement was performed to yield the final 5.1 Å resolution map. All resolution evaluation was performed based on the gold-standard criterion of the FSC coefficient at 0.143[52].

## Model building and refinement

For A3H-VCBCC complex, atomic models derived from crystal structures of human A3H (PDB ID: 5W45) and VCBCC complex (PDB ID: 4N9F) were docked into the cryo-EM map using UCSF Chimera[53]. Cullin-repeats 2 and 3 of CUL5 (residues 152–320) were not visible or only visible at low isosurface threshold level in the cryo-EM map, therefore truncated from the model. RNA strands bound to A3H were built de novo. A3H RNA binding loops 1 and 7 were rebuilt due to the large conformational change from the apo-form structure. The model was refined with the phenix.real_space_refine module in Phenix, with secondary structure restraints and geometry restraints[54,55]. We then manually adjusted the protein side-chain conformation and, when necessary, moved the main chains to match the density map using COOT[56]. The atomic models went through iterative cycles of real-space refinement in Phenix[57]. For the VCBCC dimer, two copies of the atomic model derived from the crystal structure of the VCBCC complex (PDB ID: 4N9F) were docked into the cryo-EM map, and the model was refined by a similar procedure to A3H-VCBCC complex. The final atomic models were validated using the comprehensive cryo-EM validation tool implemented in Phenix (Supplementary Table 1)[58]. All structural figures were generated with UCSF Chimera or ChimeraX[59].

## Reporting summary

Further information on research design is available in the Nature Portfolio Reporting Summary linked to this article.

## Data availability

The atomic models have been deposited in the PDB with accession codes: 8FVI (A3H-VCBCC complex) and 8FVJ (VCBCC complex dimer). The cryo-EM maps have been deposited in the EMDB with accession codes: EMD-29488 (A3H-VCBCC complex), EMD-29489 (VCBCC complex dimer), and EMD-29490 (A3H-VCBCCR complex). Raw electron microscopy data files have been deposited in the Electron Microscopy Public Image Archive (EMPIAR) with accession code EMPIAR-11423 (A3H-VCBCCR complex) and EMPIAR-11424 (A3H-VCBCC complex and VCBCC complex dimer). Source data are provided with this paper.

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

## Acknowledgements

Electron microscopy data were collected at the Core Center of Excellence in Nano Imaging (CNI) at USC and California NanoSystems Institute (CNSI) at UCLA, which is supported in part by grants from the National Science Foundation (DBI-1338135 and DMR-1548924). Cryo-EM data was computed at the Center for Advanced Research Computing (CARC) at USC. We thank Htet Khant, Carolyn Marks, and John Curulli for assisting with the operation and maintenance of transmission electron microscopes at CNI, Tomek Osinski for assisting with computing work at CARC, and Cornelius Gati for advice on cryo-EM sample preparation and data processing. This work was funded by NIH grant R01AI150524 to X.S.C. and R01GM071940 to Z.H.Z. F.I. was a former fellowship awardee from Nakajima Foundation.

## Author contributions

Z.H.Z. and X.S.C. supervised the project. F.I. and X.S.C. conceived the project and designed the experiments. F.I. purified the proteins and reconstituted the protein complex. F.I. and A.L.A.C. performed the cryo-EM grid screening. F.I. performed the cryo-EM data collection, image processing, and model building. F.I. and K.K. performed the functional biochemical analyses.

## Competing interests

The authors declare no competing interests.
