## [Peer Review File · Nature Communications]

Structural basis of HIV-1 Vif mediated E3 ligase targeting of host APOBEC3HREVIEWER COMMENTS

Reviewer #1 (Remarks to the Author):

I am reviewing only the technical aspects of the cryo-EM work in this manuscript.

I only have a few minor comments/requests to make in this area.

(1) Figure 2b (bottom right): I don't know whether this is an issue of the particular view shown, but the RNA model does not seem to fit the density well. Some phosphates and riboses are not centered in their densities.

(2) The authors need to show a close up of the cryo-EM map with the model fitted in for any regions where side chains are discussed. For example, for Fig.3b and c, and Fig.4b. Readers need to be able to judge how good the model building was in key regions.

(3) Extended Data Fig.4. A map of local resolution should be shown.

(4) Extended Data Fig.4b,f. The map shown does not look like a $\sim 5\text{\AA}$ map. For a largely helical structure, I can't see clear helical features. Similarly, I can't see the RNA double helix. This may be due to the local resolution in those regions, which is why a local resolution map should be shown.

Reviewer #2 (Remarks to the Author):

The manuscript entitled "Structural basis of HIV-1 Vif mediated E3 ligase targeting of host APOBEC3H" by Ito et al. reports the cryo-EM structures of human APOBEC3H (A3H) and E3 ubiquitin ligase bound to HIV-1 Vif. This is one of the most long-awaited structure data in the APOBEC3 research field. In this manuscript, the authors first explored combinations of the A3H and Vif variants based on previous literatures on HIV-1 Vif sensitivity to primate A3Hs and found N48H Vif and K97Q A3H best to reconstitute a functional complex of A3H-Vif/CBF- β /EloB/EloC/Cul5/RBX2 (VCBCCR). Then, the assembled A3H-VCBCC complex structure was successfully solved by cryo-EM at 3.2 \AA . The data showed the details on how HIV-1 Vif specifically recognizes A3H, and demonstrated that the A3H-Vif interaction mechanism was quite different from APOBEC3G-Vif interaction that was observed in a recent report of

the A3G-VCBCCR complex structure (by Dr. Gross' group). In addition, utilizing the authors' structure data, four major ubiquitination sites, K27, K50, K51 and K52, in A3H were identified. Furthermore, new extended interface of interaction between A3H and Vif/CBF- β were found. These responsible residues have not been not determined by previous mutagenesis analyses. In particular, the authors' finding on a charge-charge interaction between A3H and CBF- β is potentially important to rationalize how Vif polymorphic residue at position 48 contributes to efficiency of the A3H antagonism. All these findings are solid and unique, and bring important insights to understand how such small Vif molecule recognizes three different types of the APOBEC3 family proteins. The manuscript is well written.

In order to improve the manuscript, the authors need to amend the following points.

Major points:

- 1: Two A3H residues, E70 and K64, that have not been identified in previous reports are unique residues responsible for Vif interaction found in this study. Therefore, the authors should test Vif sensitivity of A3H mutants for these two residues and discuss about the results in text.
- 2: In Fig 1d, there were two major peaks (referred to as "1" and "2") of the MBP-A3H-VCBCCR complex(s). However, the authors do not specify which peak fraction was used for cryo-EM analysis in this study. The authors should explain about peak fraction(s) used for cryo-EM analyses and describe possible reasons about what the two peaks are in text.
- 3: In Fig 1e, the bands of "Ub3-A3H" and "Ub2-A3H" are hardly seen. Higher intensified gel image should be also placed.

Minor points:

- 4: Line 167, "D70" should be "E70".
- 5: Line 265, please confirm whether R90 is correct. It seems R93.
- 6: Line 501, "(Ito SciAdv)" should be corrected.
- 7: In Fig 5b, it is difficult to distinguish two colored regions between "A3G-RNA binding" and "A3H binding". It is better to change the color on one side region.

Reviewer #3 (Remarks to the Author):

Description:

In this manuscript, the authors determined the structure of a human A3H complex with Vif, CBFb, CUL5, ELOB and ELOC using Cryo EM at a 3.2 Å resolution. The structure reveals that there is a large interface between Vif and A3H that is distinct from the Vif-A3G and Vif-A3F interfaces. A total of 9 A3H residues and 8 Vif residues were identified as being critical for the A3H-Vif interaction. Unlike recently reported structures of A3G-Vif complexes, no interactions between RNA and Vif were involved in the interaction, although previously reported RNA-A3H interactions were present and identified in the complex. The structure provides valuable new insight into the interactions between HIV-1 Vif and A3H.

General critique:

These studies add valuable contributions to the recently described structural insights between Vif-A3F and Vif-A3G provided by these authors and others in the field. The Vif-A3H structure is largely in agreement with previous studies that generated a structural model of A3H-Vif interactions by employing extensive mutational analysis and structural docking (reference 44). Another mutational analysis study of chimpanzee A3H also provide support for the Vif-A3H interactions predicted by the structure. There is little or no discussion of these previous studies, and it is surprising that the authors did not provide additional support for the predicted Vif-A3H interactions by performing additional mutational analyses, especially of predicted Vif residues (shown in Fig. 3e). The authors should provide mutational analysis of Vif and A3H amino acids predicted by the structure, especially those that were not previously published. Predicted electrostatic interactions could be validated by testing mutations in Vif and A3H that reverse the charge in one of the two amino acids to show that the electrostatic interactions play a critical role in A3H degradation in cells and swapping the charges on both Vif and A3H should restore the interaction and A3H degradation. Such studies would validate the importance of these predicted structural interactions in infected cells.

Specific comments/critiques.

1. Vif E70 is shown to interact with A3H R93 (Fig. 3e). The authors should determine whether Vif E70R fails to induce degradation of WT A3H and whether A3H R93E is resistant to degradation by WT Vif. Degradation of A3H R93E with E70R would provide strong confirmation of the Vif-A3H interaction interface.
2. Similarly, the predicted electrostatic interactions between Vif E91 and A3H R41, and Vif D100 and A3H K92 can be tested by making mutations that reverse the charge on the Vif (E91R and D100K) and A3H residues (R41E and K92D) and determining whether swapping the charges inhibit degradation of WT A3H but restores degradation of the R41E and K92D mutant A3Hs.
3. In Fig. 1C, the NL4-3 Vif largely failed to degrade WT A3H, and the activity of the NL4-3 Vif against WT A3H was increased by the N48H mutation in Vif. In view of this result, why did the WT Vif induce completed degradation of WT A3H in Figure 4d? If the source of the Vif proteins used in Fig. 3F and 4D is different, it should be clearly shown in the Figure.

4. Figures 1c, 3f and 4d show single western blot gels and qualitative results. The authors should perform each experiment at least 3 times to verify that the results are reproducible and quantify the western blotting results and determine whether the differences observed are statistically significant.

5. On lines 287 – 289, the authors discuss that A3H residue K64 interacts with CBFb residues E76/Q77. Thus, the A3H K64E mutant is predicted to be resistant to Vif-induced degradation, whereas co-expression of CBFb mutant E76K should restore A3H degradation by WT Vif. The authors should test this prediction by generating the mutants and carrying out western blotting analysis.

Responses to Reviewers' Comments

Summary of revision: We thank the three reviewers for the support and insightful comments on our paper and the editor for being willing to publish our work! As you can see in the detailed itemized responses below, we have carried out new experiments and addressed all the comments/concerns in our revised manuscript. Major additional data include the new mutants of A3H and Vif to probe the extended interface residues identified in our cryo-EM structure. Additionally, more detailed descriptions of the cryo-EM density maps have been included to highlight the quality of the density maps and how they support our interpretation. The text and figures have been modified accordingly. Kyumin Kim, who assisted in performing the Vif-mediated degradation assay with new sets of mutants, was added as a co-author. We thank you for your careful review and suggestions, which have helped us to improve our manuscript!

Reviewer #1 (Remarks to the Author):

I am reviewing only the technical aspects of the cryo-EM work in this manuscript.

I only have a few minor comments/requests to make in this area.

(1) Figure 2b (bottom right): I don't know whether this is an issue of the particular view shown, but the RNA model does not seem to fit the density well. Some phosphates and riboses are not centered in their densities.

Ans: As shown in the local resolution map of the A3H-VCBCC complex (Extended Data Fig. 1i), the RNA is located at the outer edge of the whole complex and has a lower local resolution of up to 4 Å. The density corresponding to the dsRNA shows clear major and minor groove features and a few base-pairings near A3H are visible. Further refinement of the RNA model did not yield improved fitting to the density while maintaining good nucleic acids geometry. The RNA structure observed in our cryo-EM map is highly consistent with the previously reported crystal structures of A3H-RNA complex (references #20-22). To show the general fitting of the RNA model to the density, we added the model fitting of the overall RNA bound to A3H in Fig.2b.

(2) The authors need to show a close up of the cryo-EM map with the model fitted in for any regions where side chains are discussed. For example, for Fig.3b and c, and Fig.4b. Readers need to be able to judge how good the model building was in key regions.

Ans: A new figure highlighting the atomic model fitting of the key interface residues has been added as Extended Data Fig. 3. Cryo-EM densities corresponding to the side chains discussed throughout the manuscript are now shown in this figure. Additional density fitting of the representative structural components of the protein complex is depicted in the Extended Data Fig. 2.

The model presented in Fig.4b derives from the 3.2 Å cryo-EM structure of A3H-VCBCC complex from this study (Extended Data Fig. 1). The model was docked into the low-resolution cryo-EM map (with no side-chain features) of the A3H-VCBCCR complex (Extended Data Fig. 5f). The aim of this figure is to show the approximate location of surface-exposed clustered lysine residues (K27, K50, K51, and K52) relative to the position of CUL5-CTD/RBX2.

(3) Extended Data Fig.4. A map of local resolution should be shown.

Ans: A local resolution map of the A3H-VCBCCR complex has been added as Extended Data Fig. 5h.

(4) Extended Data Fig.4b,f. The map shown does not look like a ~5Å map. For a largely helical structure, I can't see clear helical features. Similarly, I can't see the RNA double helix. This may be due to the local resolution in those regions, which is why a local resolution map should be shown.

Ans: The local resolution map of the A3H-VCBCCR complex (new Extended Data Fig. 5h) indicated that the map has a relatively high resolution near the CUL5-CTD/RBX2 (4-5 Å), and relatively low resolution around A3H-Vif-CBF-β (5-8 Å), potentially indicating that there is relative flexibility. A hint of helical features is observed around the Cullin-repeat 2 and 3 (CR2 and CR3) and therefore a new figure panel showing the model fitting of this region has been added as Extended Data Fig. 5g. In contrast to the A3H-VCBCC complex map, the RNA double helix feature was not obvious in this map due to the low local resolution. We attempted a local refinement of this subregion but were not able to obtain an improved local map.

Reviewer #2 (Remarks to the Author):

The manuscript entitled “Structural basis of HIV-1 Vif mediated E3 ligase targeting of host APOBEC3H” by Ito et al. reports the cryo-EM structures of human APOBEC3H (A3H) and E3 ubiquitin ligase bound to HIV-1 Vif. This is one of the most long-awaited structure data in the APOBEC3 research field. In this manuscript, the authors first explored combinations of the A3H and Vif variants based on previous literatures on HIV-1 Vif sensitivity to primate A3Hs and found N48H Vif and K97Q A3H best to reconstitute a functional complex of A3H-Vif/CBF- β /EloB/EloC/Cul5/RBX2 (VCBCCR). Then, the assembled A3H-VCBCC complex structure was successfully solved by cryo-EM at 3.2 Å. The data showed the details on how HIV-1 Vif specifically recognizes A3H, and demonstrated that the A3H-Vif interaction mechanism was quite different from APOBEC3G-Vif interaction that was observed in a recent report of the A3G-VCBCCR complex structure (by Dr. Gross’ group). In addition, utilizing the authors’ structure data, four major ubiquitination sites, K27, K50, K51 and K52, in A3H were identified. Furthermore, new extended interface of interaction between A3H and Vif/CBF- β were found. These responsible residues have not been not determined by previous mutagenesis analyses. In particular, the authors’ finding on a charge-charge interaction between A3H and CBF- β is potentially important to rationalize how Vif polymorphic residue at position 48 contributes to efficiency of the A3H antagonism. All these findings are solid and unique, and bring important insights to understand how such small Vif molecule recognizes three different types of the APOBEC3 family proteins. The manuscript is well written.

Ans: Thank you for the detailed review and for the encouraging comments about this work.

In order to improve the manuscript, the authors need to amend the following points.

Major points:

1: Two A3H residues, E70 and K64, that have not been identified in previous reports are unique residues responsible for Vif interaction found in this study. Therefore, the authors should test Vif sensitivity of A3H mutants for these two residues and discuss about the results in text.

Ans: We tested the contribution of A3H E70, a new interface residue identified in our structure, to the Vif-A3H interaction. The charge-reversing mutant E70R resulted in elevated A3H level in the presence of Vif, indicating a reduced degradation by Vif (new Fig. 3g). When combined with another charge-reversing mutation in the neighboring residue D100 in A3H, the double-mutant (E70R/D100R) became fully resistant to Vif (new Fig. 3g). We then validated the roles of these electrostatic interactions by mutating Vif. Similar to A3H mutant results, R93E Vif showed reduced capacity to degrade A3H while the double mutant K92E/R93E Vif showed essentially no degradation of A3H (new Fig. 3h). These results indicate that the two electrostatic interactions are indeed critical for the A3H-Vif interaction. In addition, it appears that the A3H E70 – Vif R93 pair plays a more dominant role than A3H D100 – Vif K92 pair.

We also tested the K64E A3H to see whether the charge-reversal at this location can disrupt the interaction with CBF- β . The Vif-mediated degradation assay showed no obvious change in the A3H level for the K64E mutant, indicating that the A3H-CBF- β contact observed in our cryo-EM structure may only be playing a supportive role in the A3H-Vif complex formation. These results are shown below for your information.

2: In Fig 1d, there were two major peaks (referred to as “1” and “2”) of the MBP-A3H-VCBCCR complex(s). However, the authors do not specify which peak fraction was used for cryo-EM analysis in this study. The authors should explain about peak fraction(s) used for cryo-EM analyses and describe possible reasons about what the two peaks are in text.

Ans: We agree that it was not obvious in our original text which peak in the size-exclusion chromatography profile was used for the subsequent cryo-EM work. Therefore, the following description has been added in the text in lines 99-102: “suggesting that the A3H-VCBCC complex exists in multiple stoichiometry or oligomeric states (Fig. 1d). The complex species with the largest peak shift from the mixture of K97Q A3H and N48H Vif (equivalent to peak 1 in the bottom left SEC profile in Fig. 1d) were isolated and used for the cryo-EM study.”

3: In Fig 1e, the bands of “Ub3-A3H” and “Ub2-A3H” are hardly seen. Higher intensified gel image should be also placed.

Ans: We also noticed that the polyubiquitinated A3H product levels were intrinsically low. We have provided a higher-intensity gel image below and in the Source data file for interested readers. Due to the low signal, “Ub3-A3H” label has been removed from the Fig. 1e.

Minor points:

4: Line 167, “D70” should be “E70”.

Ans: The error has been corrected.

5: Line 265, please confirm whether R90 is correct. It seems R93.

Ans: This error has been corrected, too. The positively charged Vif residues around A3H Q97 are R90, K92, and R93. Accordingly, we revised the text line 265-266 and the labeling in Extended Data Fig. 7c.

6: Line 501, “(Ito SciAdv)” should be corrected.

Ans: The citation has been corrected to an appropriate format.

7: In Fig 5b, it is difficult to distinguish two colored regions between “A3G-RNA binding” and “A3H binding”. It is better to change the color on one side region.

Ans: We agreed. The colors used to indicate “A3G-RNA binding” and “A3H binding” were changed to darker colors to clearly show the two regions in Vif.

Reviewer #3 (Remarks to the Author):

Description:

In this manuscript, the authors determined the structure of a human A3H complex with Vif, CBFb, CUL5, ELOB and ELOC using Cryo EM at a 3.2 Å resolution. The structure reveals that there is a large interface between Vif and A3H that is distinct from the Vif-A3G and Vif-A3F interfaces. A total of 9 A3H residues and 8 Vif residues were identified as being critical for the A3H-Vif interaction. Unlike recently reported structures of A3G-Vif

complexes, no interactions between RNA and Vif were involved in the interaction, although previously reported RNA-A3H interactions were present and identified in the complex. The structure provides valuable new insight into the interactions between HIV-1 Vif and A3H.

Ans: Thank you for the detailed review and encouraging comments.

General critique:

These studies add valuable contributions to the recently described structural insights between Vif-A3F and Vif-A3G provided by these authors and others in the field. The Vif-A3H structure is largely in agreement with previous studies that generated a structural model of A3H-Vif interactions by employing extensive mutational analysis and structural docking (reference 44). Another mutational analysis study of chimpanzee A3H also provide support for the Vif-A3H interactions predicted by the structure. There is little or no discussion of these previous studies, and it is surprising that the authors did not provide additional support for the predicted Vif-A3H interactions by performing additional mutational analyses, especially of predicted Vif residues (shown in Fig. 3e). The authors should provide mutational analysis of Vif and A3H amino acids predicted by the structure, especially those that were not previously published. Predicted electrostatic interactions could be validated by testing mutations in Vif and A3H that reverse the charge in one of the two amino acids to show that the electrostatic interactions play a critical role in A3H degradation in cells and swapping the charges on both Vif and A3H should restore the interaction and A3H degradation. Such studies would validate the importance of these predicted structural interactions in infected cells.

Ans: New sets of A3H and Vif mutants are now included in the revised manuscript to probe the roles of the new residues identified in our structure (please see below). We have added more extensive discussions with regard to the previous structural models predicted by mutational analysis and docking (references #26 and #46). Our cryo-EM structure of Vif-A3H complex is in good agreement with these predicted models and our structure provided additional interface residues. Accordingly, we added the following text in lines 251-257 “For example, Ooms *et al.*⁴⁶ and Nakashima *et al.*²⁶ both demonstrated that A3H residues on $\alpha 3$ and $\alpha 4$ contribute to the sensitivity to Vif in cell-based degradation assays. Ooms *et al.* further generated a structural model by using anchor points revealed by the Vif adaptation study. For example, they demonstrated that Vif mutation E45Q can restore the degradation of Vif-resistant A3H mutant S86E. The cryo-EM structure we obtained largely agrees with the predicted model. Notably, we did observe a polar interaction between A3H S86 and Vif E45 in the structure.”

Specific comments/critiques.

1. Vif E70 is shown to interact with A3H R93 (Fig. 3e). The authors should determine whether Vif E70R fails to induce degradation of WT A3H and whether A3H R93E is resistant to degradation by WT Vif. Degradation of A3H R93E with E70R would provide strong confirmation of the Vif-A3H interaction interface.

Ans: We are answering comments 1 and 2 together below, as they are both related to testing the electrostatic interactions between A3H and Vif.

2. Similarly, the predicted electrostatic interactions between Vif E91 and A3H R41, and Vif D100 and A3H K92 can be tested by making mutations that reverse the charge on the Vif (E91R and D100K) and A3H residues (R41E and K92D) and determining whether swapping the charges inhibit degradation of WT A3H but restores degradation of the R41E and K92D mutant A3Hs.

Ans: We tested the contribution of A3H E70, a new interface residue identified in our structure, to the Vif-A3H interaction. The charge-reversing mutant E70R resulted in elevated A3H level in the presence of Vif, indicating a reduced degradation by Vif (new Fig. 3g). When combined with another charge-reversing mutation in the neighboring residue D100 in A3H, the double-mutant (E70R/D100R) became fully resistant to Vif (new Fig. 3g). We then validated the roles of these electrostatic interactions by mutating Vif. Similar to A3H mutant results, R93E Vif showed reduced capacity to degrade A3H while the double mutant K92E/R93E Vif showed essentially no degradation of A3H (new Fig. 3h). These results indicate that the two electrostatic interactions are indeed critical for the A3H-Vif interaction. In addition, it appears that the A3H E70 – Vif R93 pair plays a more dominant role than A3H D100 – Vif K92 pair. We also observed that another charge-reversing Vif mutant

R41E has largely lost the ability to degrade A3H (new Fig. 3h). These new mutational results are presented in Fig. 3g and 3h.

We further tested if swapping the charges between A3H and Vif can restore the degradation of A3H. Accordingly, we performed the degradation assay using A3H E70R - Vif R93E pair, and A3H E70R/D100R - Vif K92E/R93E pair. Interestingly, neither of the charge-swapped A3H-Vif pair showed detectible restoration of the degradation, indicating that simple charge swapping of the amino acid side chains is insufficient to recover electrostatic bonds between A3H and Vif. E91R mutation in A3H also did not show the restoration of degradation when combined with Vif R41E. The likely reason for this is that Vif R41 is not only interacting with A3H E91 but also with W90 and D94 (depicted in Fig. 3e). These results are shown below for your information.

3. In Fig. 1C, the NL4-3 Vif largely failed to degrade WT A3H, and the activity of the NL4-3 Vif against WT A3H was increased by the N48H mutation in Vif. In view of this result, why did the WT Vif induce completed degradation of WT A3H in Figure 4d? If the source of the Vif proteins used in Fig. 3F and 4D is different, it should be clearly shown in the Figure.

Ans: After determining the combination of Vif and A3H constructs that show the highest sensitivity in the Vif-mediated degradation assay in Fig. 1c, we used N48H mutant NL4-3 Vif in the rest of the degradation assays throughout the manuscript. Similarly, K97Q mutant A3H was used as a background sequence to add mutations to A3H in the degradation assays presented in Figs. 3f, 3g, and 4d. We agree that our original figure labeling in Fig. 3f and 4d was confusing. Therefore, Vif and A3H labels in Figs. 3f and 4d have been modified to indicate their background sequences used in these assays. Figure legends have also been revised to indicate the background sequences used to avoid confusion.

4. Figures 1c, 3f and 4d show single western blot gels and qualitative results. The authors should perform each experiment at least 3 times to verify that the results are reproducible and quantify the western blotting results and determine whether the differences observed are statistically significant.

Ans: We have performed the quantification of the western blots from three independent experiments in Figs 1c, 3f, 3g, 3h, and 4d to confirm the reproducibility and to show that the observed differences in A3H levels are statistically significant. Bar charts with individual data points have been added next to the corresponding figures. All the raw data including gel images are now included in the Source data file.

5. On lines 287 – 289, the authors discuss that A3H residue K64 interacts with CBFβ residues E76/Q77. Thus, the A3H K64E mutant is predicted to be resistant to Vif-induced degradation, whereas co-expression of CBFβ mutant E76K should restore A3H degradation by WT Vif. The authors should test this prediction by generating the mutants and carrying out western blotting analysis.

Ans: We tested the K64E A3H to see whether the charge-reversal at this location can disrupt the interaction with CBF-β. The Vif-mediated degradation assay showed no obvious change in the A3H level for K64E mutant, indicating that the A3H-CBF-β contact observed in our cryo-EM structure may only be playing a supportive role in the A3H-Vif complex formation. The assay result is shown below for your information. The Vif-mediated

degradation assay relies on the endogenous CBF- β (and the components of CUL5 E3 Ub ligase) expressed in the cells, and it is difficult to test the mutations in CBF- β in our assay system.

In **summary**, thank you again for the careful review and the valuable suggestions. We have taken the efforts in order to fully address the issues raised and revised the paper and figures accordingly.

REVIEWERS' COMMENTS

Reviewer #1 (Remarks to the Author):

The authors have addressed all my concerns.

Reviewer #2 (Remarks to the Author):

I think that the authors answered to my all questions and revised the manuscript perfectly.

Reviewer #3 (Remarks to the Author):

The authors have addressed all of the concerns raised and performed charge-swapping mutational analysis of A3H E70R and Vif R93E, and A3 E70R/D100R – Vif K92E/R93E. Unfortunately, the pairs of charge swapping mutations did not restore degradation of A3H, which would have provided strong confirmation of the structure. Nevertheless, I agree with the authors that other interactions or minor differences in the orientation of the amino acid side chains may prevent restoration of the interaction and A3H degradation. My only recommendation is that the data shown in the rebuttal letter response to Reviewer 3, specific comment 2 be included in the final paper, perhaps as a supplemental figure, so that the readers are aware of the result.

Responses to Reviewers' Comments (2nd revision)

Reviewer #1 (Remarks to the Author):

The authors have addressed all my concerns.

Reviewer #2 (Remarks to the Author):

I think that the authors answered to my all questions and revised the manuscript perfectly.

Reviewer #3 (Remarks to the Author):

The authors have addressed all of the concerns raised and performed charge-swapping mutational analysis of A3H E70R and Vif R93E, and A3 E70R/D100R – Vif K92E/R93E. Unfortunately, the pairs of charge swapping mutations did not restore degradation of A3H, which would have provided strong confirmation of the structure. Nevertheless, I agree with the authors that other interactions or minor differences in the orientation of the amino acid side chains may prevent restoration of the interaction and A3H degradation. My only recommendation is that the data shown in the rebuttal letter response to Reviewer 3, specific comment 2 be included in the final paper, perhaps as a supplemental figure, so that the readers are aware of the result.

Ans: We agree that the results of the charge-swapping mutants should be included in a supplementary figure to inform readers. Therefore, the data used in the rebuttal describing the analyses of the A3H E70R - Vif R93E pair and A3H E70R/D100R - Vif K92E/R93E pair have been added as a new Supplementary Fig. 5. The following description has been added in the text lines 167-170 "Of note, swapping the charges between A3H and Vif by using A3H E70R - Vif R93E pair and A3H E70R/D100R - Vif K92E/R93E pair did not restore the degradation of A3H, indicating that other interactions or minor differences in the orientation of the amino acid side chains may prevent the restoration of the interaction between A3H and Vif (Supplementary Fig. 5)."